# A Yolo-Based Semantic Segmentation Model for Solar Photovoltaic Panel Identification

**DOI:** 10.3390/s26010075

**Published:** 2025-12-22

**Authors:** Jiandong Zhang, Daqing Chen, Bo Li, Zhanfang Zhao, Huibo Bi, Perry Xiao

**Affiliations:** 1School of Engineering and Design, London South Bank University, 103 Borough Road, London SE1 0AA, UK; zhangj18@lsbu.ac.uk (J.Z.); chend@lsbu.ac.uk (D.C.); zhanfang.zhao@lsbu.ac.uk (Z.Z.); 2Department of Applied Mathematics, The Hong Kong Polytechnic University, Hong Kong 999077, China; cnroselearn@gmail.com; 3College of Metropolitan Transportation, Beijing University of Technology, Beijing 100124, China; huibobi@bjut.edu.cn

**Keywords:** solar panel detection, energy generation evaluation, deep learning, YOLO-based model, attention mechanism

## Abstract

The global shift towards renewable energy is increasingly driven by the need to reduce carbon emissions and address urban energy demands sustainably. Solar power, as an accessible and efficient energy source, offers substantial potential for integration within urban environments. However, there remains a lack of a comprehensive evaluation framework for accurately predicting the energy generation of urban solar panel installations. Therefore, in this study, we develop a YOLO-based semantic segmentation framework to estimate the energy generation potential of existing solar panels in a city-scale fashion and use the Elephant andCastle area of London city as a case study. The results demonstrate that the proposed model can detect and segment solar panels in complex urban environments with an accuracy of 98.32%, and the total area of solar panels in the designated area is 127.75 m^2^.

## 1. Introduction

The increasing shortage of fossil fuel resources, coupled with rising energy demands and pollution, has accelerated the global transition toward renewable energy sources. Unlike traditional energy options, renewable sources—such as solar, wind, biomass, and geothermal—offer a flexible, environmentally friendly approach to energy generation and consumption, attracting substantial interest from both industry and academia [1,2]. Among these alternatives, solar energy stands out as one of the most practical options, thanks to its high efficiency, ease of deployment, and safety [3].

Solar power has been experiencing significant growth worldwide, with China emerging as a global leader in both the adoption and production of solar photovoltaic (PV) technology [4]. In recent years, China has made remarkable strides in expanding its solar power capacity. According to the International Renewable Energy Agency (IRENA), China’s installed solar PV capacity increased from 130.25 GW in 2017 to 392.61 GW in 2022 [5]. Meanwhile, cities consume 78% of global energy [6], and urban environments with their extensive built spaces present an ideal setting for photovoltaic (PV) installations. Building “solar cities” thus holds considerable potential to reduce carbon emissions and promote carbon-neutral urban infrastructures [7]. Despite these advantages, effectively harnessing solar energy at a city scale faces significant challenges. The most critical and complex challenge is to accurately assess the energy generation potential of large-scale solar deployment in urban areas [8], which could benefit the energy planning and sustainable development of urban areas, as well as the optimization of smart grid. To address the above challenges, this paper utilizes satellite imagery to assess the solar energy generation potential at the city scale with the proposal of a Yolo-based evaluation model.

The remainder of the paper is structured as follows: We will first review the related work in Section 2, followed by the introduction of the satellite imagery-based solar panel area estimation framework in Section 3. Then, we present the Yolo-based semantic segmentation model in a detailed manner in Section 4. After that, we use the Elephant and Castle area of London city as a case study for experiments and discuss the results in Section 5. Finally, we draw conclusions in Section 6.

## 2. Related Work

### 2.1. Solar Panel-Based Energy Harvesting

Owing to its cleanliness and usability, the effort of introducing solar power into urban areas has become an emerging trend in academia. For instance, in [9], the economic and environmental analysis of installing a photovoltaic noise barrier along a metro line in China has been discussed; different metrics such as net present value [10], pay-back period, and energy pay-back time are employed to assess the economic viability, profitability, and environmental benefits of installing a photovoltaic system; the on-site collected data show that the use of the photovoltaic noise barrier can generate approximately 5000 KWh/year with notable social benefit of reducing pollution. The work in [11] investigates the potential of installing roof-mounted solar photovoltaic devices; a geographic information system (GIS)-based approach is designed to compute the usable roof area and the potential installed capacity, and the potential electricity generation is estimated, and the approximate annual electricity production for Vasteras, Sweden, on pitched roofs is 801 GWh. Similarly, the study in [12] designs an analysis framework to assess the potential of employing photovoltaic systems on buildings to offset residential electricity consumption in Shenzhen, China, and the simulation results show that the total photovoltaic energy production exceeds 88% of the total local electricity demand. The research in [13] designs a solar city framework with the aid of 3D urban surface modeling and solar photovoltaic planning; the analysis shows that solar photovoltaic plays a key role in urban decarbonization.

### 2.2. Solar Panel Segmentation Algorithms

Since accurately estimating the power capacity of solar panels is essential for effective energy planning, grid integration, and maximizing renewable energy utilization, much research has been dedicated to assessing the installed capacity of solar panels with satellite or aerial images. The Feature Pyramid Network (FPN) [14] has been widely adopted for its ability to capture multi-scale features, making it effective for semantic and instance segmentation tasks. Leveraging this capability, FPN has also become a key component in modern YOLO (You Only Look Once) models, enabling hierarchical feature fusion that enhances object localization and segmentation performance. For example, the research in [15] analyzes the characteristics and challenges of computer vision-based rooftop photovoltaic panel image segmentation with deep learning-based segmentation and clustering approaches; a comprehensive methodology is designed involving statistical, pixel-resolution, and visual feature analysis across multi-resolution datasets; the results identify that class imbalance, resolution thresholds, and homogeneous texture and heterogeneous color are the key factors that affect the performance of semantic segmentation. The work in [16] focuses on detecting defects in solar panels to ensure their safety and efficiency during long-term operation; a YOLOv5-based deep learning model is constructed and trained using an enhanced dataset with diversified solar panel defect types, combined with data augmentation and transfer learning techniques; experimental results show that the model achieves 96.5% accuracy in identifying six categories of defects, demonstrating high reliability and applicability for real-world photovoltaic inspection tasks. Similarly, the study in [17] detects defects in solar panels with various computer vision-based models; YOLO v5, v9, v10, and v11 are trained and tested in both thermal and optical datasets; the results show that YOLOv11-X achieves the best performance among all YOLO and classical machine learning models. Instead of focusing on rooftop photovoltaic panels, the work in [18] proposes a semantic segmentation model, namely PVNet, to extract industrial-scale photovoltaic panels composed of several arrays from high-resolution remote sensing imagery; the proposed PVNet method consists of a coarse prediction module to identify PV panel areas and a fine optimization module to improve boundary precision through residual refinement; the experimental results show that the proposed algorithm outperforms classical segmentation algorithms in key metrics such as precision and Intersection over Union (IoU). The study in [19] designs a hyperspectral solar segmentation network to enhance the low-quality solar panel satellite imagery for higher segmentation efficiency; by using histogram equalization to enhance image quality, hyperspectral synthetic decomposition to divide images into 31 spectral bands for feature selection, and replacing the conventional U-net layers with Chebyshev transformation layers, the results show that the proposed model surpasses state-of-the-art CNN (Convolutional Neural Network) and transformer models in accuracy, efficiency, and scalability. The study in [20] develops a dual-branch UNet-based framework for extracting photovoltaic panels from heterogeneous GF-2 and Sentinel-2 imagery; by incorporating spatial attention, channel attention, and a feature fusion module to address cross-sensor spatial–spectral inconsistencies, the proposed PV-UNet demonstrates clear advantages over mainstream CNN- and transformer-based segmentation models. The work in [21] similarly develops a detail-oriented photovoltaic segmentation network inspired by the DeepLabv3+ architecture; by enhancing multi-scale context aggregation through atrous convolution, refining boundary details with an edge-preserving module, and employing a decoder optimized for fine-grained spatial reconstruction, the proposed PV-Refiner achieves more precise delineation of solar panel regions and consistently outperforms baseline DeepLabV3+, UNet, and other commonly used segmentation models. Likewise, the work in [22] introduces a refined PSPNet for semantic segmentation of polarimetric SAR imagery in agricultural regions; through multilevel feature fusion, a polarimetric channel attention mechanism, and an edge-aware loss function, the method achieves more consistent spatial representations and surpasses widely used segmentation architectures such as PSPNet, UNet, DeepLabV3+, MP-ResNet, and DAFN. Similarly, the study in [23] proposes a transformer-based photovoltaic extraction framework tailored for heterogeneous remote sensing imagery; by leveraging global self-attention to model long-range spectral–spatial dependencies, together with multi-source normalization and contrast-enhanced feature refinement, the method effectively mitigates inter-sensor variations and exhibits stronger robustness than conventional CNN-based segmentation networks. Despite the substantial progress in solar panel segmentation, several research gaps remain to be addressed:**Lack of an end-to-end framework for converting segmented solar panel regions into accurate area estimations.** Most existing approaches focus solely on pixel-level segmentation, without providing a complete pipeline that links segmentation outputs to physical rooftop area measurements required for installed capacity estimation.**Sensitivity to real-world imaging conditions.** Shadows, occlusions, roof material diversity, panel aging, dust accumulation, and low-contrast imagery still degrade segmentation robustness across both CNN-based and transformer-based models.**Weak boundary preservation.** Thin and irregular rooftop panel edges are easily blurred or fragmented, as many current architectures do not explicitly enforce edge consistency, resulting in incomplete or imprecise boundary delineation.

To address these gaps, the present study develops a complete end-to-end pipeline for solar panel area estimation and enhances segmentation robustness by integrating attention mechanisms that improve feature discrimination under challenging imaging conditions and strengthen boundary preservation.

## 3. Solar Panel Area Estimation Framework

Based on the above literature review, there remains a lack of models capable of directly estimating the area of solar panels from satellite images in a pipeline manner. To address this gap, this section proposes an area estimation framework that trains a predictive model to learn the relationship between satellite imagery and the corresponding solar panel areas. The workflows can be divided into a training process and a predicting process. The diagram of the training workflow is shown in Figure 1 and is composed of three steps: (1) preprocessing the solar panel image dataset, (2) training a solar panel segmentation model, and (3) testing the model. The workflow of the predicting process is illustrated in Figure 2 and consists of four steps: (1) downloading satellite images, (2) converting them into image tiles, (3) performing semantic segmentation operations, and (4) calculating the area of solar panels.

In the training process, we first use a mixed dataset to train and validate our model; the dataset integrates a dataset collected from Jiangsu Province, China [24], and a dataset collected in France [25]. The dataset contains 22112 aerial images including original images and labeled solar panel positions (see Figure 3). In this research, we use 70% of the data for training, 20% of the data for validation during the training process to prevent overfitting, and the remaining 10% of the data for testing.

On the other hand, in the predicting process, we use self-collected satellite imagery of the Elephant and Castle area in London city as a case study to test the proposed model. Firstly, we download the digital map images via the Environmental Systems Research Institute (ESRI) World Imagery service interface [26]; then, we cut each digital map image into image tiles; after that, we utilize our model to conduct solar panel segmentation and calculate the total area of all detected panels.

In addition to estimating the physical surface area of photovoltaic installations, the segmentation output of the proposed framework serves as the foundation for downstream solar power forecasting. Specifically, the extracted pixel-level masks are converted into real-world PV surface area using the Web Mercator ground-resolution model, enabling the calculation of the installed PV capacity based on standard power density assumptions. This estimated capacity is subsequently integrated into an AI-aided solar power generation forecasting model, which predicts the capacity utilization factor (CUF)—defined as the ratio of the actual energy output to the theoretical maximum output under full-capacity operation. The forecasting model leverages multi-dimensional weather variables (including DHI, DNI, GHI, temperature, humidity, wind speed, and cloud cover), real-time solar radiation data, and historical CUF measurements to accurately predict CUF using a Long Short-Term Memory (LSTM) network. Finally, the expected solar electricity generation is obtained as the product of the predicted CUF and the estimated PV capacity. This two-stage area-to-energy workflow ensures that the segmented spatial information can be seamlessly transformed into actionable energy estimates, enhancing the practical applicability of the proposed framework for urban solar energy assessment and planning.

## 4. Yolo-Based Semantic Segmentation Model

To compute the area of existing solar panels, the most critical step is to recognize and segment these solar panels. To this end, we develop a YOLO11x-seg based [27,28] objective segmentation model with an extra attention module (see Figure 4) to compute the area of solar panels within the designated area. The YOLOv11x-seg model is an extension of the YOLO object detection framework, designed specifically for semantic segmentation tasks. It integrates the efficiency and speed of the YOLO architecture with advanced segmentation capabilities, enabling it to detect and segment objects simultaneously in real time. The model consists of a backbone module, a neck module, and a segmentation head module. The backbone module is a pretrained convolutional neural network that extracts feature maps from the input image; it is composed of several customized convolutional layers and C3k2 layers, an SPPF layer, and a C2PSA layer. In our modified architecture, the backbone further incorporates SimAM, a parameter-free neuron-level attention mechanism that adjusts the importance of spatial activations without introducing additional learnable parameters. The SimAM block is integrated at the end of the backbone to strengthen spatial feature discrimination, enabling the model to more effectively refine and highlight solar panel regions. The neck module is where features from the backbone are refined and passed to the segmentation heads. It consists of two repeated sequences of upsampling, concatenation, and C3k2 layers and two repeated sequences of convolutional, concatenation, and C3k2 layers. The segmentation head is used to perform pixel-wise classification to output a mask where each pixel corresponds to the solar panel class; it takes multi-scale feature maps from different pyramid layers and outputs detection predictions (bounding box, objectness, and class scores) along with a 32-dimensional mask coefficient vector for each instance. In parallel, a prototype network generates 256 prototype masks, which are linearly combined with the instance-specific coefficients to produce the final instance masks. Table 1 summarizes the major training hyper-parameters and architectural settings adopted for the proposed model.

For an input image $I\in R^{C_{in}\times H\times W}$, the designed framework first employs convolutional layers to extract features of solar panels at different scales. For the convolutional layers, it applies a k×k convolution with stride s=3 and padding p=2 to produce $C_{o}$ output channels, followed by batch normalization to stabilize training and SiLU activation to increase nonlinearity. In our implementation, the YOLO-based downsampling convolutions adopt the standard configuration of kernel size k=3, stride s=2, and padding p=1, consistent with the default YOLO design for spatial reduction. The detailed equations of the convolutional layer are shown below:(1)$$Z_{n,i,j,c_{o}}=\sum_{c=1}^{C_{\mathrm{in}}}\sum_{u=0}^{k_{h}-1}\sum_{v=0}^{k_{w}-1}W_{o,c,u,v}X_{n,s_{h}i-p_{h}+d_{h}u,s_{w}j-p_{w}+d_{w}v,c}+b_{o}$$
where $Z_{n,i,j,c_{o}}\in R^{N\times C_{\mathrm{out}}\times H^{\prime }\times W^{\prime }}$ denotes the output feature value corresponding to the *n*-th input image at spatial position (i,j) of output channel $c_{o}$, with $i=0,\ldots ,H^{\prime }-1$ and $j=0,\ldots ,W^{\prime }-1$. $W_{o,c,u,v}\in R^{C_{\mathrm{out}}\times C_{\mathrm{in}}\times k_{h}\times k_{w}}$ represents the convolution kernel weights, and $b_{o}\in R^{C_{\mathrm{out}}}$ is the bias term. Indices $o=1,\ldots ,C_{\mathrm{out}}$ and $c=1,\ldots ,C_{\mathrm{in}}$ refer to output and input channels, while (u,v) with $u=0,\ldots ,k_{h}-1$ and $v=0,\ldots ,k_{w}-1$ denote kernel offsets. Terms $s_{i},s_{j}$, $p_{i},p_{j}$, and $d_{u},d_{v}$ correspond to stride, padding, and dilation in the vertical and horizontal directions, respectively. The input tensor *I* is indexed at positions $\left(s_{i}i-p_{i}+d_{u}u,\;s_{j}j-p_{j}+d_{v}v\right)$, and values outside the valid range are treated as zeros (zero padding).

The output spatial dimensions $H^{\prime }$ and $W^{\prime }$ are given by the following:$$H^{\prime }=\left\lfloor \frac{H+2p_{i}-d_{u}\;\left(k_{h}-1\right)-1}{s_{i}}\right\rfloor +1\;W^{\prime }=\left\lfloor \frac{W+2p_{j}-d_{v}\;\left(k_{w}-1\right)-1}{s_{j}}\right\rfloor +1$$

To stabilize training and accelerate convergence, the convolution output is normalized and re-scaled using batch normalization:(2)$${\hat{Z}}_{n,i,j,c_{o}}=\gamma_{o}\cdot \frac{Z_{n,i,j,c_{o}}-\mu_{o}}{\sqrt{\sigma_{o}^{2}+\varepsilon }}+\beta_{o}$$
where $Z_{n,i,j,c_{o}}$ denotes the convolution output of the *n*-th input feature map at spatial position (i,j) of channel $c_{o}$, $\mu_{o}$ and $\sigma_{o}^{2}$ are the mean and variance of channel $c_{o}$ computed over the mini-batch, $\gamma_{o},\beta_{o}\in R^{C_{\mathrm{out}}}$ are learnable scale and shift parameters, and ε is a small constant for numerical stability.

To introduce nonlinearity and enhance feature representation, a gating mechanism based on the sigmoid function is applied:(3)$$Y_{n,i,j,c_{o}}={\hat{Z}}_{n,i,j,c_{o}}\cdot \sigma \left({\hat{Z}}_{n,i,j,c_{o}}\right)\;\sigma \left(x\right)=\frac{1}{1+e^{-x}}$$
where ${\hat{Z}}_{n,i,j,c_{o}}$ is the batch-normalized feature value, σ(·) denotes the sigmoid activation, and $Y_{n,i,j,c_{o}}$ represents the final output feature map after activation.

The convolutional layer is stacked in tandem with a C3k2 block. The C3k2 block is a variant of the cross-stage partial (CSP) structure designed to enhance feature representation while maintaining computational efficiency. In this block, the input feature map is first split along the channel dimension into two branches. The first branch, known as the shortcut branch, applies only a 1×1 convolution to preserve part of the input information with minimal transformation. The second branch, referred to as the residual branch, processes the features through two consecutive bottleneck blocks, where each bottleneck consists of a 1×1 convolution for channel reduction, a 3×3 convolution for spatial feature extraction, and a residual connection to retain original information. The outputs from these two branches are then concatenated along the channel axis and passed through a final 1×1 convolution to fuse the features, producing the output of the C3k2 block. By combining lightweight computation with deeper feature extraction, the C3k2 structure provides an effective balance between efficiency and representational power. The detailed equations of the C3k2 block are shown below.(4)$$C3k2\left(X\right)={\mathrm{Conv}}_{1\times 1}\left(\mathrm{Concat}\left({\mathrm{Conv}}_{1\times 1}\left(X\right),\;B^{2}\left(X\right)\right)\right)$$
where $X=Y_{n,i,j,c_{o}}$ is the input feature map, ${\mathrm{Conv}}_{1\times 1}\left(X\right)$ represents the shortcut branch, $B^{2}\left(X\right)=B\left(B\left(X\right)\right)$ denotes two consecutive bottleneck transformations on the residual branch, Concat concatenates the two branches along the channel axis, and the outer ${\mathrm{Conv}}_{1\times 1}$ fuses the concatenated features into the final block output. To further enhance the representation capacity, we adopt the bottleneck structure defined as follows:(5)$$B\left(X\right)=X+{\mathrm{Conv}}_{3\times 3}\left({\mathrm{Conv}}_{1\times 1}\left(X\right)\right)$$
where $X=Y_{n,i,j,c_{o}}$ is the input feature map, ${\mathrm{Conv}}_{1\times 1}$ reduces the channel dimension, ${\mathrm{Conv}}_{3\times 3}$ extracts spatial features, and the residual addition preserves the original information while enhancing feature representation.

After feature extraction by the convolutional layers, the network employs an SPPF module followed by a C2PSA module. The SPPF (Spatial Pyramid Pooling—Fast) module applies successive pooling operations with different receptive fields and concatenates the results, enabling the network to capture multi-scale spatial context efficiently. The C2PSA (cross-stage partial with parallel self-attention) module then integrates cross-stage partial connections with both channel and spatial attention mechanisms, which enhances long-range dependencies and improves the discriminative ability of the fused features.(6)$$SPPF\left(X\right)={\mathrm{Conv}}_{1\times 1}\left(\mathrm{Concat}(X,Y_{1},Y_{2},Y_{3})\right)$$
where *X* is the output feature map of Equation (Equation 4), $Y_{1}={\mathrm{MaxPool}}_{5\times 5}\left(X\right)$, $Y_{2}={\mathrm{MaxPool}}_{5\times 5}\left(Y_{1}\right)$, and $Y_{3}={\mathrm{MaxPool}}_{5\times 5}\left(Y_{2}\right)$. Here, Concat denotes channel-wise concatenation of multi-scale pooled features, and ${\mathrm{Conv}}_{1\times 1}$ is used for dimensionality reduction and fusion.(7)$$C2PSA\left(X\right)={\mathrm{Conv}}_{1\times 1}\left(\mathrm{Concat}(X_{1},A)\right)$$
where *X* is the output feature map of Equation (Equation 6), $X_{1},X_{2}$ are two splits of *X* along the channel dimension, and *A* is the attention-enhanced feature computed from $X_{2}$.(8)$$A=\alpha \cdot \mathrm{CA}\left(X_{2}\right)+\beta \cdot \mathrm{SA}\left(X_{2}\right)$$
where CA denotes channel attention, SA denotes spatial attention, and α,β are balancing coefficients. The concatenated features are finally fused by a 1×1 convolution to form the block output.

After passing through the SPPF and C2PSA modules, the feature map is further refined by a SimAM (Simple Attention Module) block [29]. The SimAM module introduces a parameter-free attention mechanism that adaptively emphasizes informative regions while suppressing less relevant responses, thereby improving the representational power of the extracted features without additional learnable parameters. Together, the SPPF, C2PSA, and SimAM modules enhance the features in a complementary manner: SPPF captures multi-scale spatial context, C2PSA strengthens long-range dependencies via channel and spatial attention, and SimAM highlights fine-grained salient information, resulting in richer and more discriminative feature representations.(9)$$E\left(x_{n,c,y,x}\right)={\left(x_{n,c,y,x}-\mu_{n,c,y,x}\right)}^{2}+\lambda \cdot \sigma_{n,c,y,x}^{2}$$(10)$$Y_{n,c,y,x}=\sigma \;\left(-\frac{E(x_{n,c,y,x})}{\tau }\right)\cdot x_{n,c,y,x}$$
where $x_{n,c,y,x}$ denotes the feature value at spatial position (y,x) of channel *c* in the *n*-th input feature map. Terms $\mu_{n,c,y,x}$ and $\sigma_{n,c,y,x}^{2}$ are the mean and variance of the local region centered at $x_{n,c,y,x}$, λ is a balancing coefficient, τ is a temperature scaling factor, σ(·) is the sigmoid function, and $Y_{n,c,y,x}$ is the refined feature after applying SimAM attention.

Following the SimAM block, the network adopts a neck module based on a Feature Pyramid Network (FPN) structure. The neck is designed to fuse multi-scale features from different backbone stages and to enhance both high-level semantics and low-level spatial details. It consists of two repeated operations, each composed of upsampling, feature concatenation with the corresponding backbone output, and a C3k2 block (already defined in Equation (Equation 4)). This hierarchical design ensures that shallow and deep features are integrated, enabling robust detection of objects at different scales.(11)$$F^{\left(1\right)}=C3k2\left(\mathrm{Concat}\left(\mathrm{Upsample}\left(F_{L_{9}}\right),\;F_{L_{5}}\right)\right)$$(12)$$F^{\left(2\right)}=C3k2\left(\mathrm{Concat}\left(\mathrm{Upsample}\left(F^{\left(1\right)}\right),\;F_{L_{3}}\right)\right)$$
where $F_{L_{3}},F_{L_{5}}$, and $F_{L_{9}}$ denote the feature maps extracted from backbone stages of third, fifth, and ninth layers with downsampling strides of 32, 16, and 8, respectively, Upsample(·) is a 2× upsampling operation (nearest-neighbor interpolation as defined in Equation (Equation 13), Concat(·) denotes channel-wise concatenation of two feature maps as defined in Equation (Equation 14), and C3k2(·) is the transformation block defined in Equation (Equation 4). Here, $F^{\left(1\right)}$ and $F^{\left(2\right)}$ are the fused feature maps generated at the first and second stages of the neck, respectively.(13)$$Y_{c,i,j}=X_{c,\;\lfloor i/r\rfloor ,\;\lfloor j/r\rfloor }$$
where $X\in R^{C\times H\times W}$ is the input feature map, *r* is the upsampling factor (typically r=2), $Y\in R^{C\times (rH)\times (rW)}$ is the upsampled feature map, and (i,j) indexes the spatial coordinates of the output, while (⌊i/r⌋,⌊j/r⌋) denotes the corresponding location in the input feature map.(14)$$Y=\mathrm{Concat}(X^{\left(1\right)},X^{\left(2\right)})$$
where $X^{\left(1\right)}\in R^{C_{1}\times H\times W}$ and $X^{\left(2\right)}\in R^{C_{2}\times H\times W}$ are two input feature maps with the same spatial size, and the output $Y\in R^{(C_{1}+C_{2})\times H\times W}$ is obtained by concatenating $X^{\left(1\right)}$ and $X^{\left(2\right)}$ along the channel dimension.

The segmentation head is designed to generate instance-specific masks in parallel with bounding box and class predictions. It consists of two branches: a prototype branch that produces a fixed set of global prototype masks shared across all instances, and a coefficient branch that predicts a coefficient vector for each detected instance. The final instance masks are obtained by linearly combining the prototype masks with the predicted coefficients, followed by sigmoid activation and optional cropping within the bounding boxes. All parameters are optimized end-to-end using a segmentation loss, which ensures that prototypes, projection weights, and coefficients are learned jointly during training. The overall formulation of the segmentation head is expressed as follows:(15)$$M_{n,h,w}=\sigma \;\left(\sum_{r=1}^{d}c_{n,r}\;\left(\sum_{c=1}^{C_{p}}W_{p}\left[r,c\right]\;P_{c,h,w}+b_{r}\right)\right)$$
where $P=G(F;\theta_{p})$ denotes the prototype masks generated from the input feature maps *F* by function G with parameters $\theta_{p}$; $W_{p}$ is the projection matrix, and $W_{p}\left[r,c\right]$ denotes its (r,c)-th element mapping prototype channel *c* to the reduced channel *r*; $b\in R^{d}$ is the bias vector with $b_{r}$ its *r*-th component; $c_{n}=H\left(F;\theta_{h}\right)\in R^{d}$ is the coefficient vector for the *n*-th instance predicted by function H with parameters $\theta_{h}$; σ(·) is the sigmoid activation; and $M_{n,h,w}$ is the predicted mask probability at spatial position (h,w) for instance *n*. Here, r=1,…,d indexes the reduced (projected) channels, $c=1,\ldots ,C_{p}$ indexes’ prototype channels, and we set $C_{p}=256$ in our experiments.

The reduced prototype representation is obtained through a 1×1 convolution as follows:(16)$$P_{r,r,h,w}=\sum_{c=1}^{C_{p}}W_{p}\left[r,c\right]\;P_{c,h,w}+b_{r}\;r=1,\ldots ,d\;\left(d=32\right)$$
where $W_{p}\in R^{d\times C_{p}}$ is the projection matrix, $b\in R^{d}$ is the bias vector, with $b_{r}$ denoting its *r*-th component, and $P_{r,r,h,w}$ denotes the reduced prototype at spatial position (h,w) for channel *r*.

The final instance mask is generated by linearly combining the reduced prototypes with the predicted coefficients as follows:(17)$$M_{n,h,w}=\sigma \;\left(\sum_{r=1}^{d}c_{n,r}\;P_{r,r,h,w}\right)$$
where σ(·) denotes the sigmoid activation, and $M_{n,h,w}$ is the soft mask probability at spatial position (h,w) for instance *n*.

The segmentation loss function used to supervise mask learning is defined as follows:(18)$$L_{\mathrm{seg}}=\frac{1}{N}\sum_{n=1}^{N}\left(\lambda_{\mathrm{bce}}\cdot \mathrm{BCE}\left(M_{n},M_{n}^{\mathrm{gt}}\right)+\lambda_{\mathrm{dice}}\cdot \mathrm{DiceLoss}\left(M_{n},M_{n}^{\mathrm{gt}}\right)\right)$$
where *N* is the number of instances, $M_{n}^{\mathrm{gt}}$ is the ground-truth mask for instance *n*, BCE(·) and DiceLoss(·) are the binary cross-entropy and Dice loss functions, respectively, and $\lambda_{\mathrm{bce}},\;\lambda_{\mathrm{dice}}$ are their balancing weights.

During training, the segmentation loss $L_{\mathrm{seg}}$ is minimized with respect to all parameters of the segmentation head, and the gradients are propagated through the entire architecture in a unified manner. Specifically, the error signals flow from the final mask predictions $M_{n,h,w}$ back to the instance coefficients $c_{n}$ and their prediction branch $\theta_{h}$, while simultaneously being transmitted to the projection weights $W_{p}$ and the prototype representations $P_{c,h,w}$, and are further propagated to update the parameters of the prototype generator $\theta_{p}$. In this way, the prototype masks, the projection matrix, and the instance coefficients are jointly optimized, enabling the model to learn coherent and discriminative mask representations in an end-to-end fashion.

To explicitly describe how the segmentation loss propagates gradients through the segmentation head, we decompose the overall gradient flow as follows:(19)$$\frac{\partial L_{\mathrm{seg}}}{\partial \{\theta_{p},W_{p},\theta_{h},c\}}=\frac{\partial L_{\mathrm{seg}}}{\partial M}\cdot \frac{\partial M}{\partial \{P_{c,h,w},W_{p},c\}}\cdot \frac{\partial \{P_{c,h,w},c\}}{\partial \{\theta_{p},\theta_{h}\}}$$
where $M=\{M_{n,h,w}\}$ is the set of predicted masks for all instances, $P_{c,h,w}$ are the prototype masks generated by the prototype branch with parameters $\theta_{p}$, $W_{p}\in R^{d\times C_{p}}$ are the projection weights, and $c=\{c_{n}\}$ are the instance-specific coefficient vectors predicted by the coefficient branch with parameters $\theta_{h}$.

The gradient with respect to the instance coefficients is given by the following:(20)$$\frac{\partial L_{\mathrm{seg}}}{\partial c_{n,r}}=\sum_{h,w}\frac{\partial L_{\mathrm{seg}}}{\partial M_{n,h,w}}\;\sigma^{\prime }\;\left(\sum_{r^{\prime }=1}^{d}c_{n,r^{\prime }}P_{r,r^{\prime },h,w}\right)\;P_{r,r,h,w}$$
where $c_{n,r}$ is the *r*-th coefficient of instance *n*, $P_{r,r,h,w}$ are the reduced prototypes, and $\sigma^{\prime }\left(z\right)=\sigma \left(z\right)\left(1-\sigma \left(z\right)\right)$ is the derivative of the sigmoid function.

Similarly, the gradient with respect to the projection weights is as follows:(21)$$\frac{\partial L_{\mathrm{seg}}}{\partial W_{p}\left[r,c\right]}=\sum_{h,w}\frac{\partial L_{\mathrm{seg}}}{\partial P_{r,r,h,w}}\;P_{c,h,w}$$
where $W_{p}\left[r,c\right]$ is the element of the projection matrix mapping prototype channel *c* to the reduced channel *r*, $P_{c,h,w}$ is the *c*-th prototype at spatial position (h,w), and $\frac{\partial L_{\mathrm{seg}}}{\partial P_{r,r,h,w}}$ is obtained by backpropagation from Equation (Equation 20).

The gradient with respect to the prototypes is as follows:(22)$$\frac{\partial L_{\mathrm{seg}}}{\partial P(c,h,w)}=\sum_{r=1}^{d}\frac{\partial L_{\mathrm{seg}}}{\partial P_{\mathrm{red}}\left(r,h,w\right)}\;W_{p}\left[r,c\right]$$
where P(c,h,w) is the prototype value at channel *c* and location (h,w), and the gradient accumulates from all reduced channels via $W_{p}$.

The gradient with respect to the prototype branch parameters is expressed as follows:(23)$$\frac{\partial L_{\mathrm{seg}}}{\partial \theta_{p}}=\sum_{c=1}^{C_{p}}\sum_{h=1}^{H}\sum_{w=1}^{W}\frac{\partial L_{\mathrm{seg}}}{\partial P_{c,h,w}}\cdot \frac{\partial P_{c,h,w}}{\partial \theta_{p}}$$
where $\theta_{p}$ are the learnable parameters of the prototype generator $G(F;\theta_{p})$, and $P_{c,h,w}$ denotes the prototype feature value at channel *c* and spatial position (h,w).

Similarly, the gradient with respect to the coefficient branch parameters is as follows:(24)$$\frac{\partial L_{\mathrm{seg}}}{\partial \theta_{h}}=\sum_{n=1}^{N}\sum_{r=1}^{d}\frac{\partial L_{\mathrm{seg}}}{\partial c_{n,r}}\cdot \frac{\partial c_{n,r}}{\partial \theta_{h}}$$
where $\theta_{h}$ are the learnable parameters of the coefficient prediction head $H(F;\theta_{h})$, and $c_{n,r}$ is the *r*-th coefficient for instance *n*.

The total physical area of detected solar panels is estimated by converting the number of foreground pixels in the segmentation mask into square meters. This conversion relies on the ground resolution of the Web Mercator projection, expressed as *meters per pixel (mpp),* which depends on the zoom level *z* and the latitude φ of the image center. When solar panels span across multiple tiled images, each visible portion is segmented independently, and the final area is obtained by summing pixel-level mask counts over all tiles. This aggregation strategy prevents both double-counting and underestimation in boundary regions, ensuring accurate estimation even when panels intersect tile borders. The final area is computed as the product of the pixel count and the squared ground resolution:(25)$$A=N_{\mathrm{pix}}\cdot {\left(\mathrm{mpp}(\varphi ,z)\right)}^{2}$$
where *A* denotes the estimated physical area in square meters (m^2^), $N_{\mathrm{pix}}$ represents the number of foreground pixels (solar panel pixels) in the segmentation mask, and mpp(φ,z) is the ground resolution in meters per pixel at latitude φ and zoom level *z*.

The ground resolution is defined as follows:(26)$$\mathrm{mpp}\left(\varphi ,z\right)=\frac{c_{z}\cdot \cos\left(\varphi \right)}{2^{z}}\cdot \frac{p_{z}}{T}$$
where φ is the latitude of the image center (in radians when used in cos(φ)), *z* is the integer zoom level of the tile, $p_{z}$ is the pixel dimension of the retrieved tile image (in our treatment, $p_{z}=256$), *T* is the tile size in pixels (typically T=256), and $c_{z}$ = 156,543.03392 m/pixel is the resolution factor at the equator for zoom level z=19 in the Web Mercator projection.

## 5. Experiments and Discussion

To evaluate the effectiveness of the proposed model, the original YOLO11-seg model was also employed for comparison purposes. Both models were trained and fine-tuned on a mixed dataset, which consists of 22,112 satellite images in total. The dataset was split into three subsets to ensure robust training and evaluation: 70% (15,478 images) for training, 20% (4422 images) for validation, and the remaining 10% (2212 images) for testing. All experiments were conducted on a workstation equipped with an NVIDIA GeForce RTX 4070 Ti GPU (12 GB VRAM), running Python 3.11 with PyTorch 2.9.1 and CUDA 13.0 acceleration. The proposed model contains 114 layers with 4.38 million parameters and 19.0 GFLOPs after fusion. During inference, the system achieves an average runtime of 1.7 ms per image, indicating that the network is highly efficient and suitable for large-scale satellite image segmentation tasks.

Figure 5 illustrates the training dynamics of the proposed model, including loss curves and evaluation metrics across 50 epochs. From the first row, it can be observed that the training losses for bounding box regression, segmentation, classification, and distribution focal loss steadily decrease, indicating stable convergence of the model. Similarly, the validation losses shown in the second row exhibit a consistent downward trend, confirming that the model generalizes well without obvious overfitting. The precision and recall curves for both bounding box detection (B) and mask segmentation (M) demonstrate continuous improvement, with precision reaching over 0.85 and recall stabilizing around 0.70 by the end of training. Furthermore, the mean Average Precision (mAP) at IoU threshold 0.5 (mAP50) reaches approximately 0.78 for bounding boxes and 0.72 for masks, while the stricter mAP50–95 metric reaches 0.60 and 0.52, respectively. These results suggest that the proposed model achieves robust detection and segmentation performance, with a favorable balance between precision and recall. The smoothness and stability of all curves further confirm the effectiveness of the adopted network design and training strategy.

Figure 6 presents sample qualitative results of the proposed segmentation model on the validation dataset. The model successfully detects and segments rooftop solar panels in high-resolution satellite imagery, with the predicted masks shown in blue overlaid on the original images. The results demonstrate that the model is able to accurately capture both the shape and spatial distribution of solar panels across varying roof orientations, scales, and background contexts. Although some overlaps between bounding boxes and masks can be observed due to densely arranged panels, the overall predictions align well with the ground-truth annotations, confirming the robustness of the model in handling complex urban environments.

Figure 7 presents the precision–confidence curve for the solar panel segmentation task based on mask predictions. The precision steadily improves as the confidence threshold increases, reaching 1.0 at approximately 0.952 confidence. This trend indicates that the model is highly reliable in distinguishing true positives when stricter confidence criteria are applied, effectively eliminating false detections. Compared with lower confidence thresholds, where precision values fluctuate, the curve demonstrates that the model achieves robust segmentation performance when confidence is set above 0.5. Such characteristics are essential for practical deployment, where ensuring accurate mask-level detection is critical to avoid false segmentation of non-solar regions.

To comprehensively evaluate the segmentation quality of the proposed model, we employ three widely used metrics: Intersection over Union (IoU), F1-Score, and Pixel Accuracy. These metrics capture complementary aspects of segmentation quality: IoU measures the spatial overlap between predicted and ground-truth masks, F1-Score reflects the balance between precision and recall, and Pixel Accuracy quantifies global correctness across the entire image. Together, they provide a more complete and reliable assessment of model performance than any single metric alone. IoU is defined as$$\mathrm{IoU}=\frac{TP}{TP+FP+FN}$$
which measures the overlap between predicted masks and the ground truth. Although IoU is known to be sensitive to small objects and class imbalance, rooftop PV panels in our task form large, continuous regions where IoU provides a stable and meaningful measure of mask alignment. To mitigate the known limitations of IoU, we additionally report F1-Score and Pixel Accuracy, which together offer a more complete assessment of segmentation performance.

The F1-Score is expressed as$$F1=\frac{2TP}{2TP+FP+FN}$$
providing a balance between precision and recall. Pixel Accuracy is defined as$$\mathrm{Accuracy}=\frac{TP+TN}{TP+TN+FP+FN}$$
which evaluates the overall proportion of correctly classified pixels.

As illustrated in Figure 8, the proposed model achieves consistently high performance, with mean IoU, F1-Score, and Accuracy reaching 0.8564, 0.9117, and 0.9952, respectively. In comparison, the baseline YOLO11-Seg model attains lower values of 0.8174, 0.8779, and 0.9663. This demonstrates that our proposed model provides a clear improvement across all three evaluation metrics. Furthermore, the area comparison in Figure 9 demonstrates that the predicted segmentation areas closely follow the variations of the ground-truth areas, confirming that our method not only ensures pixel-level accuracy but also preserves global structural consistency.

To further examine the robustness of the proposed segmentation model in shaded urban environments, we conduct an additional experiment on the PV01 rooftop subset of the Jiangsu multi-resolution dataset for photovoltaic panel segmentation [24]. PV01 consists of 0.1 m UAV images of distributed rooftop PV installations in dense urban areas, where many rooftops are partially covered by cast shadows from neighboring buildings or trees. Such shadows often make PV modules visually similar to surrounding roof materials and dark background regions, and have been reported to reduce segmentation accuracy for small-scale rooftop systems. On the PV01 test split, our proposed model achieves segmentation performance with mean IoU of 0.9156, F1-Score of 0.9549, and Mean Accuracy of 0.9802 (see Figure 10). These results suggest that the proposed method maintains good robustness in partially shaded urban scenes, although extreme shadow occlusions remain challenging and will be addressed in our future work.

As a case study, we conducted experiments in a designated study area located in the Elephant and Castle district of London, UK, as illustrated in Figure 11. High-resolution satellite imagery of this area was obtained using the ESRI world imagery service. The selected region is bounded by the geographical coordinates (51.5020, −0.1097) and (51.4914, −0.0898), covering the Elephant and Castle area, and was divided into image tiles of size 256×256. The tiled satellite images were processed through the proposed segmentation framework, and the real-world area was computed by converting pixel counts using the meter-per-pixel factor (see Equations (Equation 25) and (Equation 26)) derived from the zoom level and the (x, y) tile indices stored in each image filename. These indices uniquely determine the tile’s geographic position and allow accurate estimation of ground resolution. After segmentation, all detected rooftop PV regions were manually inspected and corrected to ensure the reliability of the final measurement. Following this refined procedure, the total rooftop solar panel area in the study region was determined to be 127.75 m^2^.

## 6. Conclusions

This study presented a YOLO-based semantic segmentation framework for solar photovoltaic (PV) panel detection and area estimation in urban environments. By introducing the enhanced architectural module SimAM, the proposed model significantly improved segmentation performance compared with the baseline YOLO11-Seg. Experimental evaluation on the Elephant and Castle area of London demonstrated that the proposed model achieved a mean IoU of 0.8564, a mean F1-Score of 0.9117, and a Mean Accuracy of 0.9952, surpassing the YOLO11-Seg baseline ( 0.8174, 0.8779, and 0.9663, respectively). Moreover, the predicted solar panel areas closely matched the ground truth, with an average relative difference of less than 2.5%, thereby confirming the reliability of our approach not only at the pixel level but also in area estimation. In addition to these technical improvements, the accurate quantification of rooftop PV areas provides an important foundation for urban energy-flow optimization, supporting data-driven scheduling of distributed renewable resources and contributing to long-term reductions in carbon emissions. Our future research will also explore enhancing model robustness under real-world degradation factors such as shadows, soiling, panel aging, and adverse weather conditions—limitations inherent in current satellite PV datasets. Beyond this, we aim to leverage generative adversarial networks (GANs) [30] to generate synthetic satellite images and simulate various urban scenarios, and explore hybrid models that combine LSTM [31] with other machine learning techniques such as convolutional neural networks or attention mechanisms [32]. Moreover, we will explore the possibility of introducing spiking neural networks for efficient solar panel detection and segmentation [33,34,35,36], and the optimization of energy flows across solar power resources with operational models [2,37].

## Figures and Tables

**Figure 1 sensors-26-00075-f001:**
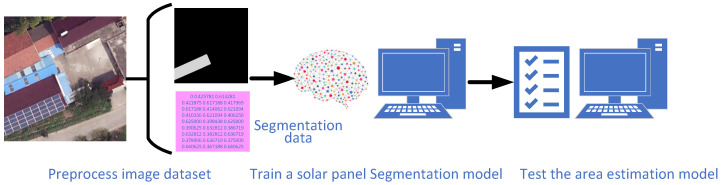
The diagram of the training workflow of the solar panel area estimation framework.

**Figure 2 sensors-26-00075-f002:**
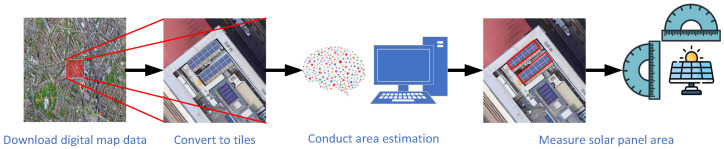
The diagram of the computational workflow of the solar panel area estimation.

**Figure 3 sensors-26-00075-f003:**
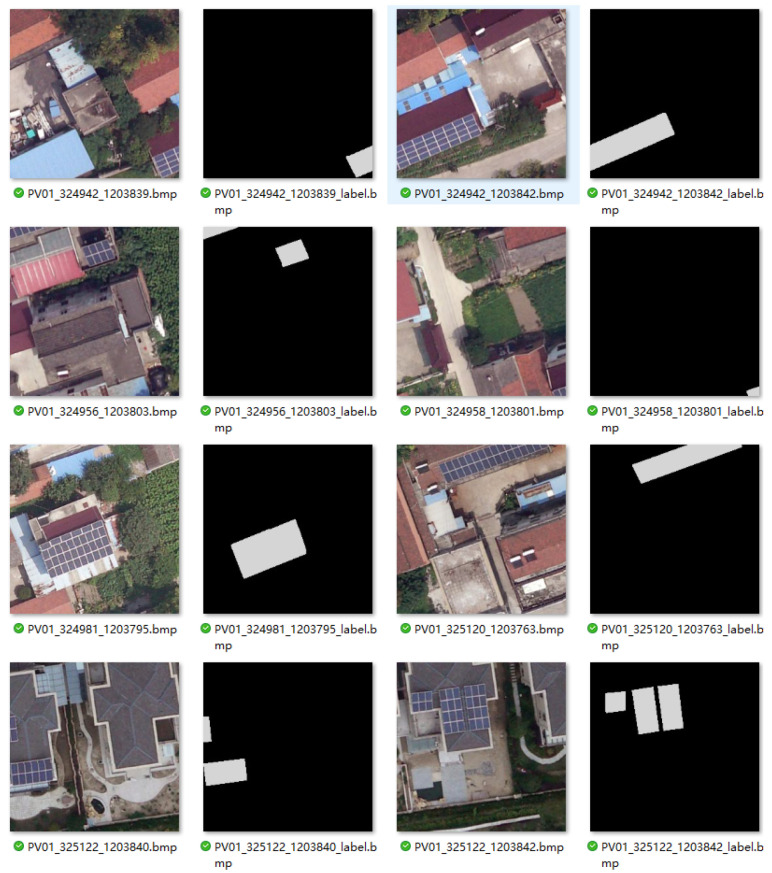
Demonstration of the multi-resolution satellite imagery dataset used for the photovoltaic panel segmentation task.

**Figure 4 sensors-26-00075-f004:**
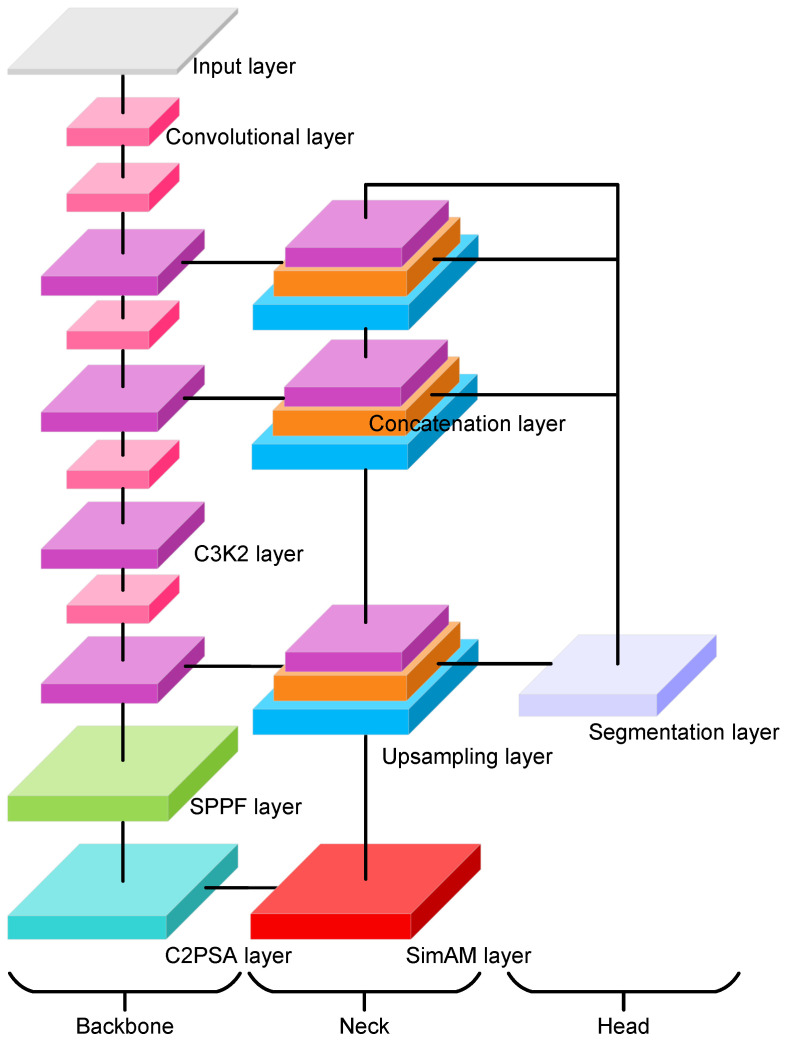
The proposed semantic segmentation framework based on YOLOv11x-seg and SimAM.

**Figure 5 sensors-26-00075-f005:**
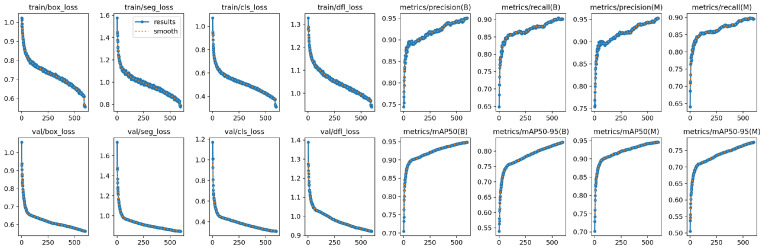
The training dynamics of the proposed model.

**Figure 6 sensors-26-00075-f006:**
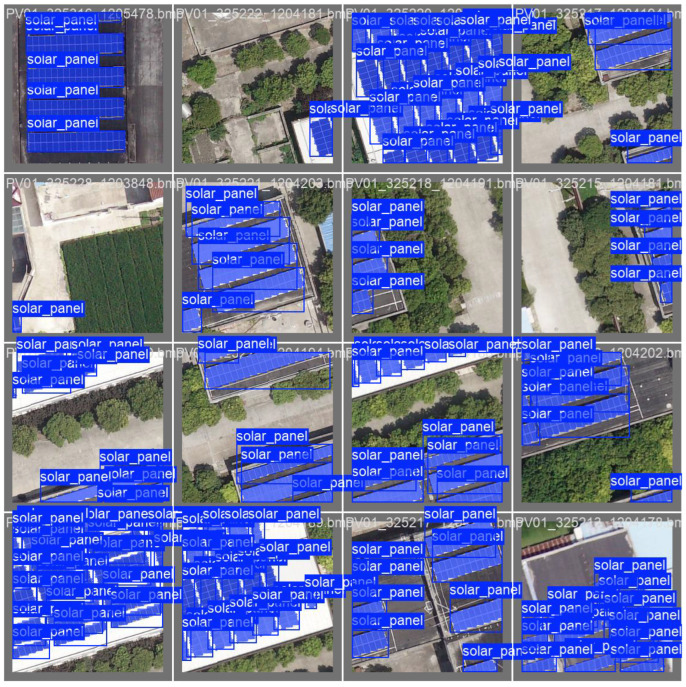
The sample results of the proposed segmentation model in complex urban environments.

**Figure 7 sensors-26-00075-f007:**
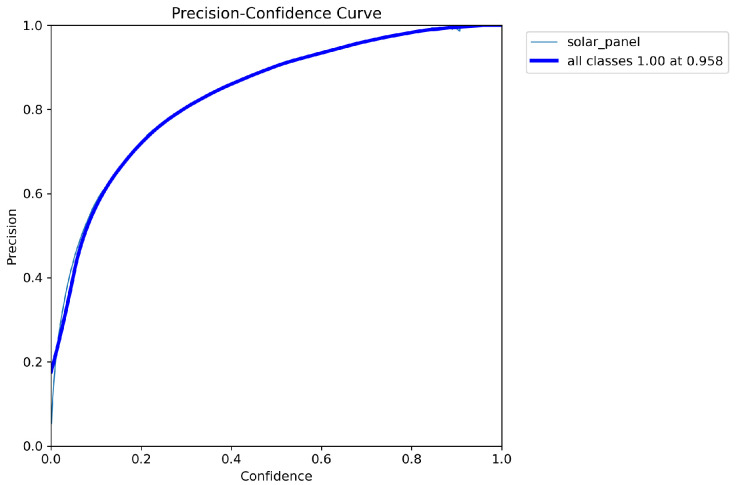
The precision–confidence curve for the solar panel segmentation task.

**Figure 8 sensors-26-00075-f008:**
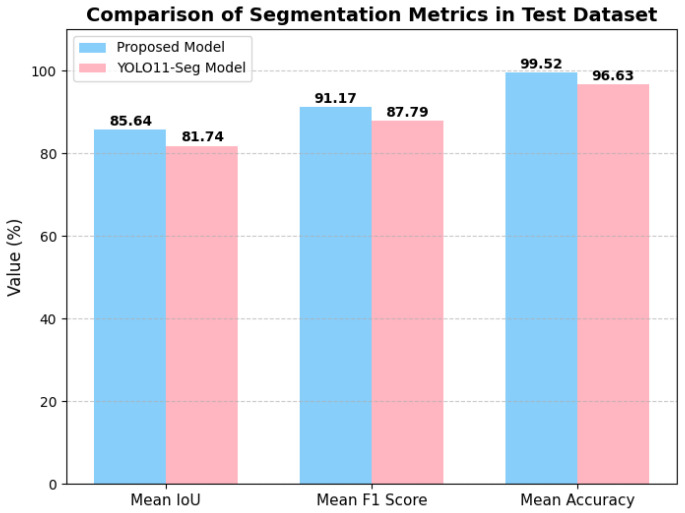
Comparison of segmentation accuracy among the various models.

**Figure 9 sensors-26-00075-f009:**
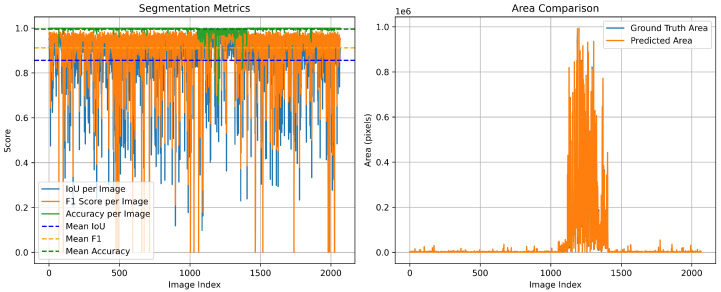
Segmentation metrics and area comparison of the proposed model in the test dataset.

**Figure 10 sensors-26-00075-f010:**
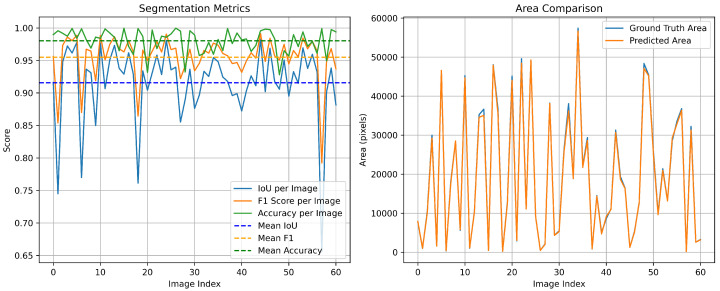
Segmentation metrics and area comparison of the proposed model in the Jiangsu multi-resolution PVO1 test dataset (63 images).

**Figure 11 sensors-26-00075-f011:**
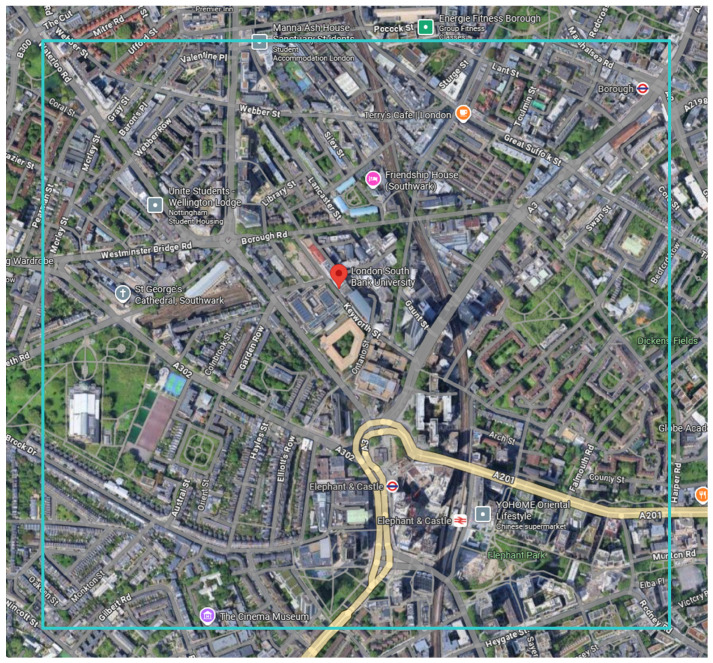
The designated area for the evaluation of the proposed model.

**Table 1 sensors-26-00075-t001:** Hyper-parameter settings for the proposed model.

Parameter	Value
Epochs	600
Batch size	16
Input resolution for training	640×640
Input resolution for testing	640×640
Optimizer	SGD
Momentum	0.937
Initial learning rate	0.01
Learning rate scheduler	Cosine decay
Warm-up epochs	3.0
Weight decay	0.0005
Base model	YOLO11-seg

## Data Availability

The data presented in this study are available on request from the corresponding author.

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
