# Peer review of "A Yolo-Based Semantic Segmentation Model for Solar Photovoltaic Panel Identification"

_sensors, 2025, doi:10.3390/s26010075_

Round 1
Reviewer 1 Report
Comments and Suggestions for Authors
The authors proposed an innovative deep learning model for semantic segmentation, ie. YOLO-based semantic segmentation framework, for identifying solar PV panels. After reading the paper, I have the following comments:
General comment: The research objective is clear and meaningful. The proposed method is appropriate and contains certain degree of novelty. However, I think that the literature review must be improved to cover more recent works and clearly point out the research gap. There are major problems in the model description, experiment, and discussion of result. These shortcomings must be addressed carefully to ensure that the paper meet the high quality of an international journal.
1) Literature review should include more related works about existing methods, e.g. UNet++, Feature Pyramid Network, Pyramid Scene Parsing Network, Transformer based deep learning model, and DeepLabV3; clearly discuss their accuracy, strength, and limitations.
2) Page 3, after line 95: summarize research gaps using several bullet points
3) Line 110-111: using only one dataset might be not sufficient; it is recommended that the authors rely on more datasets for training and validation, such as Rooftop photovoltaic installations in Jiangsu (China), rooftop photovoltaic installations in France, etc. Using more public and open-access datasets are very helpful to confirm the proposed method’s superiority.
4) Page 10: clearly point out the setting the model’s hyper-parameters
5) Page 11: add more details about the experiments, such as GPU capability, computational cost, inference time.
6). Section 4: It is not clear how the proposed approach deals with PV energy estimation after segmentation.
7). Detecting PV panels in urban environment can be difficult when the scene is partially shaded; how to deal with this issue? Please demonstrate the model’s performance when detecting such scenes.
8) Additionally, changes in panel appearance due to dirt, degradation, and weather conditions can reduce the detection/segmentation performance. Provide some examples for these cases and add some discussions regarding the model’s performance in these scenarios.
9) Regarding ‘total area of solar panels in the designated area is 127.75 m2 .’, how to confirm segmentation result with ground truth data?
10) Enhance the discussion regarding how the proposed method can help reduce carbon emissions and address urban energy demands sustainably.
Reviewer 2 Report
Comments and Suggestions for Authors
The paper describes experiments with a modified version of YOLO CNN to detect solar panels and calculate their area.
The paper is well organised and well written. The contents are easy to follow and in general the method proposed is coherent. The results are also in line with what could be expected.
Here are some more detailed comments to hopefully help improve the paper.
- The authors describe the yolo-based segmentation model in detail. However, the differences between the base yolo model and their modifications are not totally clear. This could be revised and improved for clarity.
- Some hyperparameters could also be specified and justified in more detail. For example, stride, padding, convolution sizes, etc. - it is not clear whether they were optimized, or why they were chosen. Even if they were randomly chosen, that could be clarified in the paper.
- Another aspect that could be referred is what happens if a solar panel is split across two or more images when images are tiled.
- Equations must be numbered and checked. IoU is notoriously wrong.
- English could be improved. There are some awkward sentences (e.g., "from industry and academic alike", or use of many "and" in a sentence).
Round 2
Reviewer 1 Report
Comments and Suggestions for Authors
I have no further comments.
Reviewer 2 Report
Comments and Suggestions for Authors
Thank you for the improved paper and clarifications. My main concerns were well addressed. I believe the paper is now clearer and easier to follow.